# Masseter Muscle Thickness Measured by Ultrasound as a Possible Link with Sarcopenia, Malnutrition and Dependence in Nursing Homes

**DOI:** 10.3390/diagnostics11091587

**Published:** 2021-08-31

**Authors:** Mikel González-Fernández, Javier Perez-Nogueras, Antonio Serrano-Oliver, Elena Torres-Anoro, Alejandro Sanz-Arque, Jose M. Arbones-Mainar, Alejandro Sanz-Paris

**Affiliations:** 1Nutrition Department, University Hospital Miguel Servet, 50007 Zaragoza, Spain; gonzalezmikel@hotmail.es; 2Geriatric Unit, Elias Martinez Nursing Home, 50007 Zaragoza, Spain; jperez@begar.es; 3Geriatric Unit, Casa Amparo Nursing Home, 50007 Zaragoza, Spain; antseroliver@gmail.com; 4Geriatric Unit, Romareda Nursing Home, 50007 Zaragoza, Spain; metorres@salud.aragon.es; 5Santa Ana Outpatient Center, 50007 Tudela, Spain; jandrosanzarque@hotmail.com; 6Instituto de Investigación Sanitaria Aragon (IIS-Aragon), 50007 Zaragoza, Spain; email@adipofat.com; 7Adipocyte and Fat Biology Laboratory (AdipoFat), Translational Research Unit, Instituto Aragones de Ciencias de la Salud (IACS), University Hospital Miguel Servet, 50007 Zaragoza, Spain; 8Centro de Investigación Biomédica en Red Fisiopatología Obesidad y Nutrición (CIBERObn), Instituto Salud Carlos III, 28029 Madrid, Spain

**Keywords:** ultrasound, masseter muscle thickness, sarcopenia, malnutrition, dependence

## Abstract

Sarcopenia is a progressive and generalized loss of skeletal muscle mass and strength. It is frequently associated with malnutrition and dependence in nursing homes. Masticatory muscle strength could be the link between sarcopenia, malnutrition and dependence. We aimed to study the relation between sarcopenia, malnutrition and dependence with masseter muscle thickness measured by ultrasound. A cross-sectional study was realized, with 464 patients from 3 public nursing homes in Zaragoza (Spain). The diagnosis of sarcopenia was assessed according to the European Working Group on Sarcopenia in Older People 2 criteria, malnutrition by the Mini Nutritional Assessment (MNA) and the Global Leadership Initiative on Malnutrition (GLIM) criteria and functional capacity by the Barhel Index and the texture diet. Masseter muscle thickness (MMT) was measured by ultrasound. The median age was 84.7 years, and 70% of the participants were women. Sarcopenia was confirmed in 39.2% of patients, malnutrition in 26.5% (risk 47.8%), total dependence in 37.9% and diet texture was modified in 44.6%. By logistic regression, once the model was adjusted for age, sex, Barthel index and texture diet, our analyses indicated that each 1 mm decrease in MMT increased the risk of sarcopenia by ~57% (OR: 0.43), the risk of malnutrition by MNA by ~63% (OR: 0.37) and the risk of malnutrition by GLIM by ~34% (OR: 0.66). We found that MMT was reduced in sarcopenic, malnourished and dependent patients, and it could be the common point of a vicious cycle between sarcopenia and malnutrition. Further studies are needed to establish causality.

## 1. Introduction

Sarcopenia is a syndrome characterized by a progressive and generalized loss of skeletal muscle mass and strength, with a risk of adverse outcomes and mortality in nursing home residents [1]. There is a wide variation in the prevalence of sarcopenia because of the characteristics of the population and the methodology. Most of the studies use the definition of the European Working Group on Sarcopenia in Older People (EWGSOP) [2]. The prevalence varies from 17% to 34% [3,4]. In a recent systematic review with more than 2685 cases, the prevalence was 41% [5]. In Spanish nursing homes, the prevalence varies from 37% [6] to 41.4% [7].

Sarcopenia and malnutrition are closely related. In a study by Beaudart et al. [8], malnutrition was associated with an approximately four-fold higher risk of developing sarcopenia during a four-year follow-up in community-dwelling patients. In another study [9], nearly two-thirds of older medical inpatients had at least one of the tissue loss syndromes: sarcopenia, frailty, cachexia and malnutrition. Moreover, 80% of malnourished patients were also sarcopenic and in addition, 53% of malnourished patients were frail.

A sarcopenia diagnosis is confirmed by the presence of low muscle quantity. Muscle mass can be estimated in different ways, such as via magnetic resonance imaging (MRI), computed tomography (CT) and dual-energy X-ray absorptiometry (DEXA). However, these imaging tests are not useful in everyday practice because of the lack of portability, disponibility, the costs and the radiation. Bioimpedance (BIA) is a more widely available technique, but it is an indirect technique [10]. Ultrasound has been widely used lately for the assessment of muscle mass [11].

Sarcopenia is a global process, and for this reason, the masticatory muscles are also affected. The masseter, temporalis and the lateral and medial pterygoid muscles are the masticatory muscles of the head and neck [12]. The masseter muscle is the strongest and most important masticatory muscle [13]. Moreover, it is a thick, superficial and easy to locate muscle in an uncovered area. The temporalis muscle is also superficial but thinner, and pterygoid muscles are in a deeper position, and for these reasons, they are more difficult to measure [12]. A study of Japanese community-dwelling elders observed a relationship between the thickness of the masseter muscles (MMT) and appendicular skeletal muscle mass [12]. These people with sarcopenia could have thinner masseter muscles, which would lead them to eat a texture-modified diet (TMD) (puréed diet or soft diet), ultimately provoking a vicious cycle of malnutrition and sarcopenia [14]. We have not found any studies in Western institutionalized elderly that explore the relationship between sarcopenia or malnutrition and MMT measured with ultrasound.

Although it is true that no clinical practice guidelines recommend the use of ultrasound for the diagnosis of sarcopenia, it is a widely used technique in clinical settings with great intra- and inter-observer reliability [15]. Moreover, it can be applied with minimal levels of ultrasound training, showing a good correlation with other techniques such as MRI, CT and DEXA, but with the advantages of low cost and portability (bedside use) [16].

Our hypothesis is that the measurement of MMT could be a key muscle group in the diagnosis of sarcopenia and malnutrition in institutionalized elderly. Therefore, the main objective of our study is to assess the relationship between the thickness of the masseters and the existence of sarcopenia and malnutrition. As a secondary objective, we study the cutoff points of MMT in sarcopenia and malnutrition in our population.

## 2. Materials and Methods

### 2.1. Study Design and Recruitment

A multicenter cross-sectional study was performed in three public nursing homes in Zaragoza (Spain) from February to August 2019. The inclusion criteria were residents aged older than 65 years, having resided in the nursing home for at least 6 months to ensure a stable situation and that they sign an informed consent about their participation in the study, either by the participants or their legal representatives. Of a total of 602 residents, 485 agreed to participate and, after applying the exclusion criteria, the final study population was 464 individuals. Exclusion criteria were: masseteric hypertrophy or bruxism (n = 2), tumor interventions or mandibular surgeries (n = 11), nasogastric tube (n = 6), severe intercurrent diseases (n = 8) and patients with edema (n = 4), which could alter the BIA results. Diet texture and functional capacity assessments were performed by the participants’ geriatricians, while nutritional assessment, sarcopenia and masseter muscle thickness measurements by ultrasound were performed blindly by the nutrition unit physicians.

### 2.2. Sarcopenia Assessment

The prevalence of sarcopenia has been determined using the EWGSOP2 [17] criteria, which recommends four steps: (1) SARC-F as screening for clinical suspicion, (2) muscle strength by hand grip strength (HGS) for probable sarcopenia, (3) muscle quantity by bioimpedance for confirmed sarcopenia and (4) physical performance by gait speed (GS) for severe sarcopenia.

The SARC-F [18] is a 5-item questionnaire that is self-reported by patients as a screen for sarcopenia risk. Responses are based on the patient’s perception of his or her limitations in strength, walking ability, rising from a chair, stair climbing and experiences with falls. This screening tool was translated and validated in Spanish language by Parra et al. [19]. 

Muscle strength was measured by HGS because it is advised for routine use in hospital practice. HGS was measured by a Jamar Hydraulic Dynamometer Hand Strength Meter, model 5030J1 (Sammons Preston Inc., Bolingbrook, IL, USA). The participants squeezed the dynamometer using maximum isometric effort. The measurement was taken three times, following the Roberts protocol [20]. The results of HGS were classified by the EWGSOP2 sarcopenia cutoff points as <27 kg for men and <16 kg for women [21].

Muscle mass was measured by BIA (Akern BIA 101 SMT device, Florence, Italy). Electrical parameters obtained with BIA were converted to appendicular skeletal muscle mass (ASM) using the cross-validated Sergi equation [22], based on older European populations. As recommend by EWGSOP2, sarcopenia cutoff points for ASM/height^2^ (ASMI) used were <7.0 kg/m^2^ for men and <5.5 kg/m^2^ for women [23].

Physical performance was measured by GS in meters per second over a 4 m distance in usual walking speed using usual walking aids (such as canes, walkers or crutches). For simplicity, a single cutoff speed ≤ 0.8 m/s is advised by EWGSOP2 as an indicator of severe sarcopenia [24]. In our study, participants that used a wheelchair or were bedridden were directly classified as a low CST or GS and were only measured for HGS and BIA.

### 2.3. Nutritional Assessment

We use the GLIM criteria and MNA questionnaire for the nutritional assessment. We used MNA-SF as a screening test for both nutritional assessment tools. For phenotypic GLIM criteria, we assessed the following variations: unintentional weight loss > 5% within the last 6 months, body mass index (BMI) < 22 kg/m^2^ or reduction of muscle mass based on the fat-free mass index (FFMI). Muscle mass was measured with the subjects in supine position after overnight fasting by bioelectrical impedance analysis (BIA), employing the BIA 101 instrument (Akern, Italy). FFMI values lower than <17 kg/m^2^ for males or <15 kg/m^2^ in females were considered as low muscle mass according to the ESPEN consensus statement [25]. The etiologic GLIM criteria considered were either (1) reduced food intake—decreased food ingestion for at least two weeks reported by the patient or his/her caregivers and further corroborated by the institutional caretakers, or (2) acute disease inflammation (acute diseases retrieved from the patients’ clinical records during the past 3 months).

In the MNA tool, out of a maximum score of 30 points, >24 points reflects well-nourished people, 17–23.5 points reflect risk of malnutrition and those who score <17 points are considered malnourished [26].

The subjects’ current weight, height and body mass index (BMI) were measured according to standardized and recommended techniques [27,28]. Participants were measured in a room with a temperature of 25 °C and were required to take off any heavy coats. We measured weight in light clothing, with a floor-calibrated scale (SECA 880, Chino, CA 91710, USA). Patients’ height was measured using a fixed-wall stadiometer (SECA 220, USA). It was not possible to properly determine the real height in all subjects. In those cases, Chumlea´s equation to estimate height [29] was utilized. BMI was calculated as weight (kg) divided by height^2^ (m^2^).

### 2.4. Masseter Muscle Thickness Measurement

As described in a previous study [30], ultrasonographic determinations were performed with the patient in a relaxed and natural head position. Subjects were instructed to maintain minimal interocclusal contact. Recordings were performed using a 7.5 MHz linear transducer connected to an Edan DUS 60 ultrasound scanner (Edanusa, San Diego, CA, USA).

The transducer was positioned perpendicular to the external edge of the muscle, between the intertragic fissure and the oral commissure, parallel to the Frankfort plane, exerting medium pressure and applying a generous layer of transducer gel. We took measurements from the right and left masseter muscle, but we used the thickest masseter for statistical analysis. A line was drawn from the cortex of the mandibular ramus to the inner part of the fascia (Figure 1). Three measurements were taken by the same observer at 5 min intervals and the results were averaged. The person performing the ultrasound was blinded to the rest of the data.

### 2.5. Meal Forms Assessment

There were three meal forms provided at the nursing homes according to texture and food source: normal texture (equivalent to level 7 of the IDDSI), soft texture (equivalent to level 6 of the IDDSI) and puréed texture diet (equivalent to level 4 of the IDDSI) [31]. The meal form was determined by the attending doctor of the nursing homes.

### 2.6. Functional Ability Assessment

The ability to perform basic activities of daily living was assessed with the Barthel index. It classifies individuals according to different levels of functional dependence. It consists of different items, all daily life activities such as the ability to dress, wash, eat, etc., in order to determine the dependence of the subject.

Barthel index score is a 10-item ordinal scale that measures autonomy for basic activities of daily living (ADLs). Values less than 20 indicate total dependency for activities of daily living, measures between 21 and 35 indicate severe dependence, 40 to 55 moderate dependence, 60 to 95 light dependence and 100 indicates independent [32,33].

### 2.7. Statistical Analysis

The results were expressed as mean and standard deviation (SD) for quantitative parameters or as percentages of individuals for qualitative ones. Chi-square test was used to detect differences between categorical variables, and the normal distribution of continuous variables was tested by the Kolmogorov–Smirnov test. Differences in continuous variables between subgroups were analyzed by Student’s t-test or analysis of variance (ANOVA) if normally distributed. To determine the relationship between quantitative parameters, we used Spearman’s correlation coefficient (rho). In order to examine the relevant factors for MMT, multiple regression analysis (forced entry method) was used, with MMT as the dependent variable and sarcopenia diagnosis as well as malnutrition (MNA and GLIM) as the independent variables. Age, sex, Barthel index score and diet texture were also included as confounding factors among the independent variables. Multicollinearity was avoided by selecting one item when the correlation coefficient was 0.8 or more between two variables. 

Logistic multivariate analysis was conducted for the association of sex, texture of diet, nutritional status and presence of sarcopenia with MMT, and a receiver operating characteristics (ROC) curve for the estimated predictions.

SPSS version 26 was used for statistical analyses, considering a 5% significance level in all tests.

### 2.8. Ethical Aspects

The study protocol was approved by the management of the three geriatric centers and by the local ethical committee (Ethical Committee for Clinical Research of Aragon, CEIC-A, ref C.P.-C.I. PI19/135).

## 3. Results

### 3.1. Basic Characteristics

Four hundred and sixty-four residents were included in the study (70% women) (Table 1). The mean age of the cohort was 84.7 (7.7) years (range 61–108 years), with an average BMI of 24.4 kg/m^2^. The prevalence of malnutrition according to the GLIM criteria was 11.2%. According to the MNA, the prevalence of malnutrition was 26.5%, and 47.8% of the cohort was at risk of malnutrition. Of the residents, 97.6% presented some degree of dependency and only 55.4% had a diet of regular consistency. The prevalence of confirmed sarcopenia was 39.2%, similar to severe sarcopenia (38.6%), because functional impairment was very prevalent, and applying the GS test only discriminated three patients (Figure 2).

There were no differences between men and women in terms of age and BMI. However, women obtained significantly lower scores on the MNA questionnaire, and had a higher degree of dependence and a higher risk of sarcopenia. Women also required a modified texture diet more frequently and had a reduced MMT compared to men.

### 3.2. Factors Related to MMT

We found that the thickness of the masseters was significantly increased in residents on regular-textured diets compared to those who consumed soft or puréed diets (Table 2). Conversely, the presence of malnutrition, characterized by either GLIM or MNA criteria, as well as confirmed sarcopenia and severe dependence according to the Barthel index, were associated with a decreased thickness of the masseters.

Those associations were further expanded in a correlation analysis, in which we found that masseter thickness was directly correlated with ASMI (rho: 0.525, *p*: 0.0001), MNA score (rho: 0.590, *p*: 0.0001) and the Barthel index (rho 0.752, *p*: 0.0001), while negatively correlated with age (rho: −0.271, *p*: 0.0001) (Figure 3).

Subsequently, we used different logistic models to investigate whether the thickness of the masseters could predict the risk of sarcopenia and malnutrition. We found that masseter thickness was strongly associated with reduced sarcopenia and malnutrition. Thus, our analyses indicated that a 1 mm increase in MMT reduced the risk of sarcopenia by 63% (OR: 0.37), the risk of malnutrition according to the MNA by 69% and the risk of malnutrition according to the GLIM criteria by 40%. Both the effect size and the statistical significance of the risk reduction were maintained in the different models adjusted for confounding variables such as age, sex, Barthel index and diet texture (Table 3).

We then built receiver operator characteristics (ROC) curves to examine the sensitivity and specificity of MMT to predict sarcopenia and malnutrition, according to the MNA and GLIM criteria. The area under the ROC curve (AUC-ROC) for the prediction of sarcopenia was 0.843 and 0.822 for men and women, respectively (Table 4 and Table 5). Excellent discrimination power for the MMT to predict malnutrition was also observed according to MNA (0.965 and 0.897 for men and women, respectively), and somewhat reduced using the GLIM criteria (0.749 and 0.707 for men and women, respectively).

Finally, we calculated the optimal cutoff points using the Youden index (the maximum sum of sensitivity and specificity) to identify sarcopenic individuals (6.59 and 6.00 mm for men and women, respectively). Cutoffs to detect malnutrition in men were 7.21 and 6.59 mm, according to either GLIM criteria or MNA respectively, and those cutoffs appeared slightly reduced in women, at 5.78 and 6.27 mm according to either GLIM criteria or MNA, respectively.

## 4. Discussion

In our study, MMT was associated with malnutrition and sarcopenia as well as diet texture and Barthel index. Furthermore, our analyses indicated that each 1 mm decrease in MMT doubled the risk of sarcopenia and malnutrition as judged by the MNA, adjusted for confounding variables such as age, sex, Barthel index and diet texture.

There are few studies on the prevalence of sarcopenia in the very elderly population. The Newcastle 85+ Study showed sarcopenia prevalence of 21% at mean age 85.5 (0.4) years with the EWGSOP1 criteria, either at home or in an institution [34]. In the BELFRAIL study [35], in subjects aged 80 years and older living at home, the sarcopenia prevalence was only 12.5% according to the EWGSOP1 algorithm. Finally, the ULSAM study [36] in 85–89-year-old community-dwelling men showed a 21% prevalence by EWGSOP1 and 20% by EWGSOP2. We found that 39.2% of our nursing home residents (37.5% male 40% female) had confirmed sarcopenia according to the EWGSOP2 criteria. A similar prevalence was described in a recent meta-analysis (41% out of 3585 participants) and in two previous studies in the Spanish population: 37% (15% men, 46% women) [2] and 41.4% (18.6% men, 81.4% women) [36].

Selecting the most suitable method for evaluating muscle mass in a nursing home setting is challenging. It is well-accepted that the amount of muscle in the limbs represents the overall muscle mass, since most of the skeletal muscle in the body exists in the extremities [37]. However, sarcopenia is a global process and there is a loss of muscle mass and reduced muscular strength in the head and neck too. For this reason, masseter muscle could be as useful as the limb muscles to detect loss of muscle mass [38].

Determining which parameter of the masseter muscle is to be measured is also controversial, as well as the imaging technique to be performed. In this sense, ultrasonography is more portable, has a reduced cost and does not emit ionizing radiation, as compared to other imaging techniques such as scanning or magnetic resonance [16]. In our study, we observed a significant correlation between the MMT and the ASMI evaluated by BIA, which is part of the diagnostic protocol of EWGSOP2. Previous studies have shown a correlation between the masseter muscles and the ASM of young people [39]. However, little is known regarding the relation of MMT and muscle mass. Umeki et al. [12] and Yamaguchi et al. [40] found a statistically significant correlation between the thickness of the masseter and the appendicular muscle mass. Wallace [38] related masseter mass to the psoas using computed tomography. 

There is good evidence of the relationship between masticatory problems and malnutrition [41]. The masseter muscle is one of the main muscles involved in chewing and swallowing. Accordingly, Yunsun et al. [42] already posited the applicability of the masseter muscle as a nutritional biomarker. However, there are few data that establish a relationship between sarcopenia and an altered masticatory function. Moreover, these few data came from studies performed in the Asian population that described a relation between oral functional parameters and skeletal muscle mass [12] or global physical performance, such as CST and HGS [43]. Likewise, Yamaguchi found a relationship between MMT and functional capacity [44], and Gaszynska et al. also showed its relation to physical fitness [45]. Watanabe described that the risk of frailty was associated with lower occlusal force and masseter muscle thickness in 4720 Japanese elderly people. Our results in the European population showed a strong association between MMT sarcopenia, malnutrition and functional capacity, in concordance with Asiatic studies [46].

Malnutrition, sarcopenia and frailty frequently coexist in older people, mainly in nursing homes. Recently, Faxen-Irving et al. found that most residents were either (pre)frail (51%), sarcopenic (29%) or malnourished (17%), with considerable overlaps between the three conditions [47].

Finally, we observed a relationship between low MMT and TMD (puréed and soft diets). With regard to the quality of the menus offered by nursing homes, puréed diets are of special interest and concern due to the work involved in their preparation and their low nutritional content, as found in the present study. Keller et al. [48] highlighted the need to improve the nutritional quality of puréed food, Dahl et al. [49] concluded that puréed food prepared in diets in Canadian LTC homes contained inadequate levels of protein and Vucea et al. [50] reported a significant association between the consumption of a soft or puréed diet and a higher risk of malnutrition. Importantly, the energy or nutrient requirements of individuals needing a texture-modified diet do not differ from those of people of the same age and sex, except in the presence of disease [51]; therefore, a puréed menu should meet the same general dietary recommendations. The texture of the diet (e.g., puréed) was prescribed for each resident according to their needs by the physicians at the nursing homes who are usually responsible for designing these menus in Spain [52].

Given the relationship between sarcopenia, malnutrition, frailty and puréed diet with reduced MMT, we hypothesized the existence of a vicious cycle between general and localized sarcopenia in the masseter muscle, and malnutrition and frailty (Figure 4).

The muscles of the head and neck have not been well-studied globally. To the best of our knowledge, there are no data on possible cutoff points of MMT for the diagnosis of malnutrition and sarcopenia. We had found only two studies that showed cutoff points in limb muscles [53,54]. The number of participants was 44 [53] and 204 [54] community-dwelling older people, but in no case did they assess masseter muscles, and the functional capacity of the participants was good, without sarcopenia.

The main limitation of this study is the cross-sectional study design that did not allow us to clarify temporal or cause–effect relationships between sarcopenia or malnutrition and MMT. Secondly, our study excluded patients who were provided artificial enteral nutrition because of previously diagnosed malnutrition. Additionally, the number of existing teeth and functional teeth were not considered because they were related with the texture diet. This study also has some strengths: (i) its European and multicenter character, (ii) the large sample size, which allowed for the simultaneous diagnosis of sarcopenia, malnutrition and functional capacity, and (iii) that the ultrasound study was performed blindly by a radiologist who was unaware of the data from the rest of the study.

## 5. Conclusions

In the nursing home elderly population studied, we found a strong relationship between MMT and sarcopenia, malnutrition, poor diet and functional capacity. Although we cannot provide causality as it is a cross-sectional study, we have offered cutoff points for sarcopenia and malnutrition.

## Figures and Tables

**Figure 1 diagnostics-11-01587-f001:**
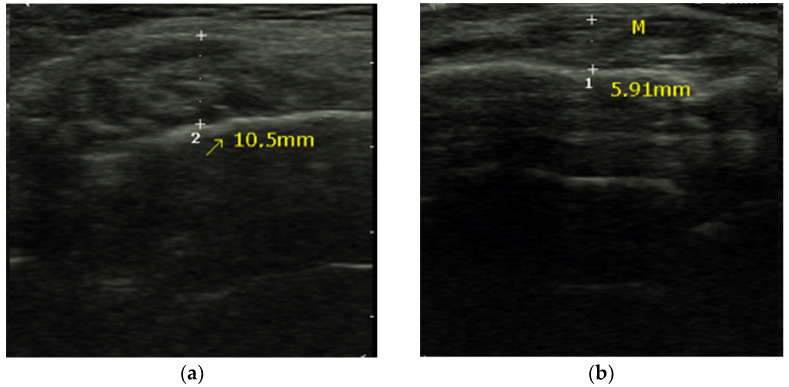
(**a**) Normal masseter muscle and (**b**) wasted masseter muscle.

**Figure 2 diagnostics-11-01587-f002:**
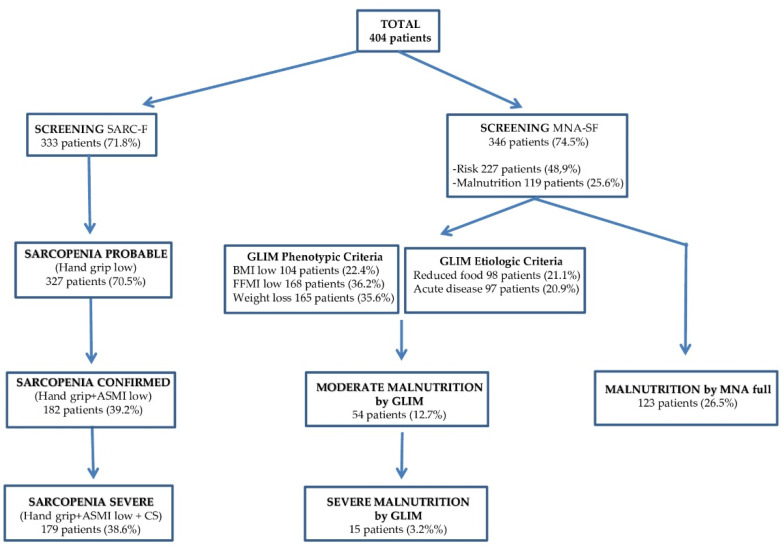
Flow chart of sarcopenia and malnutrition diagnostic factors in the total group of patients.

**Figure 3 diagnostics-11-01587-f003:**
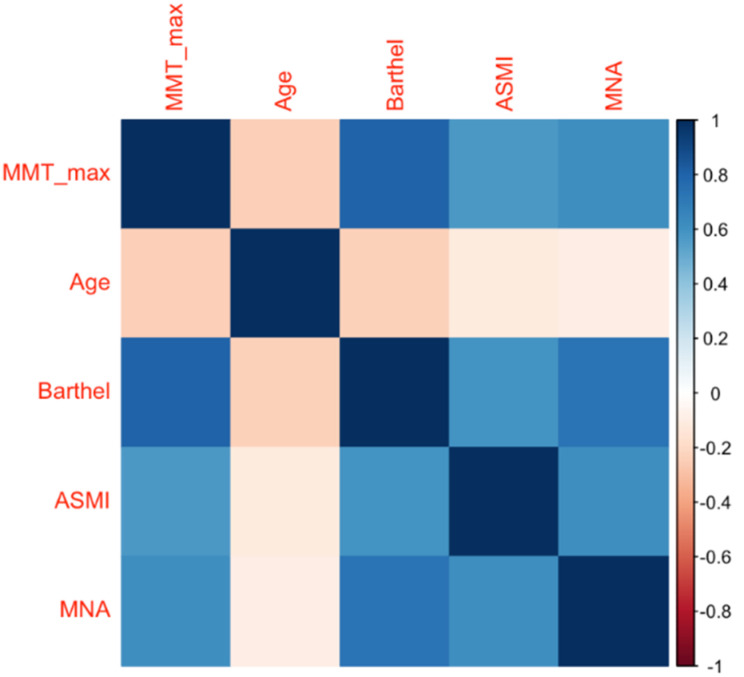
Correlogram of MMT. Correlogram illustrating the Spearman correlation coefficients between MMT and age, function (Barthel index), appendicular skeletal muscle mass index (ASMI) and nutrition (MNA). Color intensity is proportional to the correlation coefficients.

**Figure 4 diagnostics-11-01587-f004:**
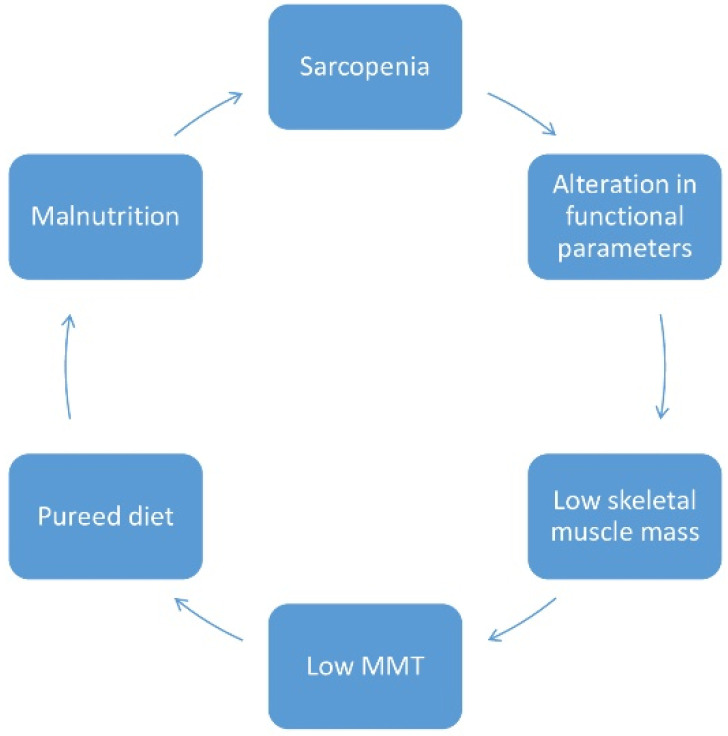
Possible vicious cycle between sarcopenia, low masseter muscle, puréed diet, malnutrition and frailty.

**Table 1 diagnostics-11-01587-t001:** Characteristics of the study group.

	All (*n* = 464)	Women (*n* = 325)	Men (*n* = 139)	*p*
Age (years)	84.7 (7.7)	85.1 (7.6)	83.7 (7.6)	0.06
BMI (Kg/m^2^)	24.4 (5.8)	24.7 (6.3)	23.8 (4.3)	0.07
MMT (mm)	6.5 (1.5)	6.2 (1.5)	7.0 (1.6)	0.0001
MNA score	19.6 (5.0)	18.9 (4.9)	20.9 (4.9)	0.0001
MNA normal (%)	25.6	20.0	38.2	0.0001
MNA risk (%)	47.8	49.7	43.8	
MNA malnutrition (%)	26.5	30.3	18.1	
GLIM normal (%)	88.8	89.7	86.8	0.2
GLIM malnutrition (%)	11.2	10.3	13.2	
Total dependence (%)	37.9	43.7	24.5	0.0001
Severe dependence (%)	13.4	15.1	9.4	
Moderate dependence (%)	15.7	16.3	14.4	
Light dependence (%)	30.6	24.0	46.0	
Independent (%)	2.4	0.9	5.8	
Regular diet (%)	55.4	51.4	64.4	0.043
Soft diet (%)	14.9	16.6	10.8	
Smooth puréed diet (%)	29.7	31.7	25.2	
Sarcopenia risk (%)	71.8	79.7	54.2	0.0001
Probable sarcopenia (%)	70.5	78.1	53.5	0.0001
Confirmed sarcopenia (%)	39.2	40.0	37.5	0.6
Severe sarcopenia (%)	38.6	39.1	37.5	0.7

BMI: body mass index; MMT: masseter muscle thickness; MNA: Mini Nutritional Assessment; GLIM: Global Leadership Initiative on Malnutrition. Continuous variables are expressed as mean (standard deviation). Categorical data are expressed as the incidence (percentage).

**Table 2 diagnostics-11-01587-t002:** Factors related to masseter muscle thickness.

Factors	MMT (mm)	Differences
Diet Texture		
Regular	7.0 (1.4)	*p*: 0.0001
Soft	6.3 (1.5)	
Puréed	5.5 (1.3)	
Malnutrition (MNA)		
Normal	7.6 (1.3)	*p*: 0.0001
Risk	6.6 (1.4)	
Confirmed malnutrition	5.1 (1.1)	
Malnutrition (GLIM)		
Normal	6.6 (1.5)	*p*: 0.0001
Confirmed malnutrition	5.5 (1.3)	
Sarcopenia		
Normal	7.3 (1.4)	*p*: 0.0001
Confirmed sarcopenia	5.5 (1.3)	
Barthel Index		
Independent	8.8 (1.5)	*p*: 0.0001
Mild	7.7 (1.3)	
Moderate	6.9 (0.8)	
Severe	6.7 (0.9)	
Total	5.1 (0.9)	

MNA: Mini Nutritional Assessment; GLIM: Global Leadership Initiative on Malnutrition; MMT: masseter muscle thickness. Continuous variables are expressed as mean (standard deviation). Categorical data are expressed as the incidence (percentage).

**Table 3 diagnostics-11-01587-t003:** Masseter muscle thickness as predictive factors of sarcopenia and malnutrition (MNA and GLIM) analyzed by logistic regression models.

	OR (RAW)	OR (Model 1)	OR (Model 2)	OR (Model 3)
Sarcopenia	0.37 (0.30–0.45, *p* < 0.001)	0.34 (0.28–0.42, *p* < 0.001)	0.42 (0.33–0.52, *p* < 0.001)	0.43 (0.34–0.54, *p* < 0.001)
Malnutrition (MNA)	0.31 (0.25–0.40, *p* < 0.001)	0.30 (0.23–0.33, *p* < 0.001)	0.34 (0.27–0.45, *p* < 0.001)	0.37 (0.29–0.49, *p* < 0.001)
Malnutrition (GLIM)	0.6 (0.48–0.75, *p* < 0.001)	0.57 (0.45–0.72, *p* < 0.001)	0.65 (0.50–0.85, *p* < 0.001)	0.66 (0.51–0.87, *p* < 0.003)

MNA: Mini Nutritional Assessment; GLIM: Global Leadership Initiative on Malnutrition; RAW: Not adjusted; MODEL 1: Adjusted for age and sex; MODEL 2: Adjusted for age, sex and Barthel index (independent/dependent); MODEL 3: Adjusted for age, sex and Barthel index (independent/dependent) and diet texture (normal/modified).

**Table 4 diagnostics-11-01587-t004:** ROC curve in men.

	Cutoff Point	Area	Significance	95% IC		Sensitivity	Specificity
				Lower Limit	Upper Limit		
Malnutrition (GLIM)	7.21 mm	0.749	0.0001	0.659	0.840	0.536	0.947
Malnutrition (MNA)	6.59 mm	0.965	0.0001	0.923	1.000	0.891	0.962
Sarcopenia	6.59 mm	0.843	0.0001	0.774	0.912	0.844	0.722

MNA: Mini Nutritional Assessment; GLIM: Global Leadership Initiative on Malnutrition.

**Table 5 diagnostics-11-01587-t005:** ROC curve in women.

	Cutoff Point	Area	Significance	95% CI		Sensitivity	Specificity
				Lower Limit	Upper Limit		
Malnutrition (GLIM)	5.78 mm	0.707	0.0001	0.607	0.808	0.662	0.727
Malnutrition (MNA)	6.27 mm	0.897	0.0001	0.844	0.950	0.862	0.867
Sarcopenia	6.00 mm	0.822	0.0001	0.775	0.869	0.776	0.766

MNA: Mini Nutritional Assessment; GLIM: Global Leadership Initiative on Malnutrition.

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
