# Peer review of "Masseter Muscle Thickness Measured by Ultrasound as a Possible Link with Sarcopenia, Malnutrition and Dependence in Nursing Homes"

_diagnostics, 2021, doi:10.3390/diagnostics11091587_

Round 1

Reviewer 1 Report

Dear Authors,

It is an interesting article, with an important topic. 

Introduction: the order of ideas must be revised (you write about sarcopenia and than ultrasound and than sarcopenia )

Methods: Who performed the different procedures?

Results: The tables must be more clean and care, the number of decimal places must be the same (tables and text) . The figures were not well defined. 

Best regards

Author Response

Please, find below our responses to the reviewers´ comments.

We would like to show our appreciation for their prompt and thorough work.

Introduction: the order of ideas must be revised (you write about sarcopenia and than ultrasound and than sarcopenia )

We agree that the logical order of ideas should be sarcopenia first and then ultrasound. Accordingly, we changed the order of the introductory paragraphs. All the paragraphs referring to sarcopenia were grouped together and the ultrasound paragraph goes in the penultimate position right before the hypothesis of the study. 

Methods: Who performed the different procedures?

Reviewer’s observation is very important for the validity of the results. For this reason, we included a “Study design and recruitment" paragraph in the MATERIAL AND METHODS section. Briefly, diet texture and functional capacity assessment was performed by their geriatricians, while nutritional assessment, sarcopenia, and masseter muscle thickness measurement by ultrasound were performed blindly by the physicians of the nutrition unit.

Results: The tables must be more clean and care, the number of decimal places must be the same (tables and text).

We regret the lack of clarity in the tables . According to the reviewer's suggestion we changed the commas to points, homogenized the number of decimals and aligned the units. We have also put the same order of the factors in tables 4 and 5.

The figures were not well defined. 

High quality figures have been uploaded in the revised version

Reviewer 2 Report

Dear Sirs, I find this paper very interesting and valid, though tiny things about it should be improved. They are as follow:

  1. line 103 - "validated"
  2. It would be valuable to add wheather the measurements were done in NHP (natural head position) or if it was not taken into account (materials and methods - 2.4)
  3. To my mind, table 1 should be situated after the 1st paragraph in the results section.

To my mind, after those changes the paper could be published

Author Response

Please, find below our responses to the reviewers´ comments.

We would like to show our appreciation for their prompt and thorough work.

line 103 - "validated".

We regret the typo. It has been corrected in the revised version.

It would be valuable to add wheather the measurements were done in NHP (natural head position) or if it was not taken into account (materials and methods - 2.4)

The reviewer’s rises a valid point that it would be informative to specify the position of the head during the scan. Consequently, we included in the material & methods section the next sentence : "As described in a previous study [30], ultrasonographic determinations were performed with the patient in a relaxed and natural head position."

To my mind, table 1 should be situated after the 1st paragraph in the results section.

The reviewer is right that table 1 should be situated after the 1st paragraph. We have modified the results section accordingly.
